# Inhomogeneous spatial distribution of the magnetic transition in an iron-rhodium thin film

C. Gatel[1], B. Warot-Fonrose[1], N. Biziere[1], L.A. Rodríguez[1,2], D. Reyes[1], R. Cours[1], M. Castiella[1] & M.J. Casanove[1]

Monitoring a magnetic state using thermal or electrical activation is mandatory for the development of new magnetic devices, for instance in heat or electrically assisted magnetic recording or room-temperature memory resistor. Compounds such as FeRh, which undergoes a magnetic transition from an antiferromagnetic state to a ferromagnetic state around 100 °C, are thus highly desirable. However, the mechanisms involved in the transition are still under debate. Here we use *in situ* heating and cooling electron holography to quantitatively map at the nanometre scale the magnetization of a cross-sectional FeRh thin film through the antiferromagnetic–ferromagnetic transition. Our results provide a direct observation of an inhomogeneous spatial distribution of the transition temperature along the growth direction. Most interestingly, a regular spacing of the ferromagnetic domains nucleated upon monitoring of the transition is also observed. Beyond these findings on the fundamental transition mechanisms, our work also brings insights for *in operando* analysis of magnetic devices.

[1] CEMES CNRS-UPR 8011, Université de Toulouse, 29 rue Jeanne Marvig, 31055 Toulouse, France. [2] Department of Physics, Universidad del Valle, A.A. 25360, Cali, Colombia. Correspondence and requests for materials should be addressed to C.G. (email: gatel@cemes.fr).

The FeRh alloy presents a remarkable and unusual magnetic transition from a low temperature antiferromagnetic state (AFM) to a high temperature ferromagnetic state (FM) close to 370 K accompanied by a 1% volume expansion[1–5]. The transition is obtained for a narrow composition range $0.48 < x < 0.56$ in the B2-ordered $\alpha'$ crystal phase of $Fe_{1-x}Rh_x$. Its main characteristics have been studied both theoretically[6–8] and experimentally using dedicated techniques such as Mössbauer[9,10], neutrons[3,11,12] and XMCD[13–18]. These remarkable features make FeRh particularly well suited for advanced magnetic devices[19] for instance in heat-assisted or even electrically assisted magnetic recording[20–22] and magnetic random access memories based on AFM spintronics[23], or for magnetocaloric materials[24]. Further developments of reliable devices including thin FeRh films or others materials presenting a magnetic transition (as MnAs[25,26]) require a full control of the magnetic state within the film, a perfect knowledge of the mechanisms involved in the transition and a deep understanding of the influence of interfaces and low-dimensionality.

Different publications reported the persistence of an FM component at room temperature or even below in ultrathin FeRh films[27,28] or at the interface between the film and the capping layers[12,14,29,30]. These observations were attributed to chemical diffusion, surface segregation or misfit strain, and often to a combination of these different effects. The role of surface or interface in the nucleation of the FM domains at the onset of the transition was also analysed experimentally[17,18,29–31]. Few studies have mapped the transition on thin films[17,18,30]. They report co-existing FM and AFM domains on the top surface and the subsequent expansion of the FM domains. In addition, the magnetic profile along the growth direction of FeRh films, for both undoped and doped films, was analysed by polarized neutron reflectometry with a sub-nm resolution[12,29,32,33]. Spectra were suitably fitted by taking into account both a top and a bottom interface regions (around 8 nm thick) surrounding the core of the film.

Here we report quantitative imaging on the local magnetization by electron holography (EH). In contrast with the previous experimental methods, this technique is highly sensitive to the magnetic signal[34–37] and information can be extracted across the entire film thickness with unrivalled spatial resolution, notably along the interfaces, compared to polarized neutron reflectometry. *In situ* EH experiments were performed on a cross-sectional 50 nm thick FeRh film to fully investigate the magnetic transition mechanism at a nanometre scale, with the objective to unravel the different effects attributed to size and interfaces. We first demonstrate that the *in situ* experiment performed on the TEM specimen, that is, on a very small area, suitably reproduces the macroscopic magnetization loop measured at the AFM–FM transition using vibrating sample magnetometry (VSM). Comparing these local and macroscopic results enables us to provide essential information on the TEM specimen, and on the effect of focused ion beam thinning (FIB) preparation. In a second part, we exploit the magnetic phase images recorded within the film upon heating and cooling cycles to map the evolution of the transition temperature from surface to interface with the substrate and quantify the magnetization changes. Our results provide insight into the transition mechanisms.

## Results

**Structural properties of the studied area.** A 50 nm FeRh film was epitaxially grown at 550 °C on a MgO (001) substrate by DC sputtering and *in situ* annealed for 6 h at 800 °C. The AFM/FM transition in this film was first studied by VSM. For EH

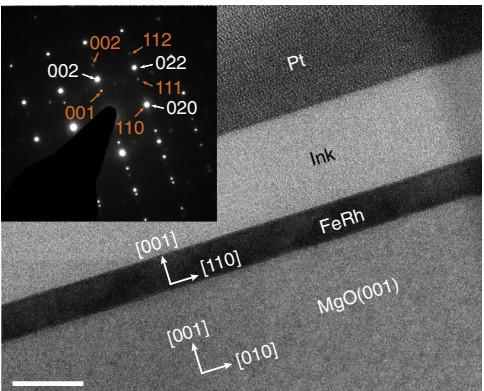

**Figure 1 | Epitaxial FeRh thin film on MgO(001) substrate.** Conventional TEM image (main panel) and diffraction pattern (inset) of the area studied by electron holography. The scale bar is 100 nm. On the diffraction pattern, indexes in orange correspond to the FeRh layer and white ones to the MgO substrate.

experiments, a cross-sectional lamella was prepared by FIB to ensure a uniform thickness crossed by the electron beam. The FeRh layer was protected by ink and platinum layers to avoid damages and charge accumulation during the thinning process. The final thickness of the lamella was measured at 90 ± 5 nm after a low energy step to minimize irradiation damages and surface amorphization (see Supplementary Fig. 1).

Figure 1 presents a TEM image of the area used for the EH study. The corresponding diffraction pattern demonstrates the epitaxial growth of the monocrystalline FeRh layer on the substrate with a 45° rotation between the FeRh bcc lattice and the MgO fcc lattice. The growth axis is [001] and the layer is observed along the FeRh [−110] axis ([100] for MgO). The appearance of the 001 reflections reveals the B2 ordered phase.

**Series of holograms and principle of the data analysis.** The basic principles of EH and the description of the full procedure for the magnetic phase shift extraction as well as for the magnetization quantification are detailed in Supplementary Notes 4 and 5. In the following, [110] and [001] directions in FeRh are chosen as $x$ axis and $y$ axis respectively. The direction parallel to the electron beam stands for the $z$ axis. Using the Aharanov–Bohm equations[38,39], the $x$ component of the induction, noted $B_x$, was extracted from a defined area of the magnetic phase image by analysing the mean slope of the magnetic phase shift $\phi_M$ along the $y$ direction:

$$\int B_x(x,y,z)\mathrm{d}z = \frac{\hbar}{e}\frac{\partial \phi_M(x,y)}{\partial y} \tag{1}$$

with $e$ the electron charge and $\hbar$ the reduced Planck constant. Measuring the lamella width $w$ crossed by the electron beam and assuming that the magnetization is equivalent to the induction inside the FeRh layer, that is, neglecting the stray fields, it is then possible to obtain the $x$ component of the magnetization, noted $\mu_O M_x$:

$$\mu_0 M_x = \frac{\int B_x(x,y,z)\mathrm{d}z}{w} \tag{2}$$

Repeating this analysis as a function of the applied temperature $T$, we then can plot $M_x = f(T)$ on a very localized area. The resulting graph is therefore strictly equivalent to the classical $M(T)$ curve obtained by magnetometry.

Prior to the EH experiment, the FeRh layer was slightly tilted away from the [−110] zone axis to avoid dynamical diffraction effects which create artefacts on the phase measurement. The

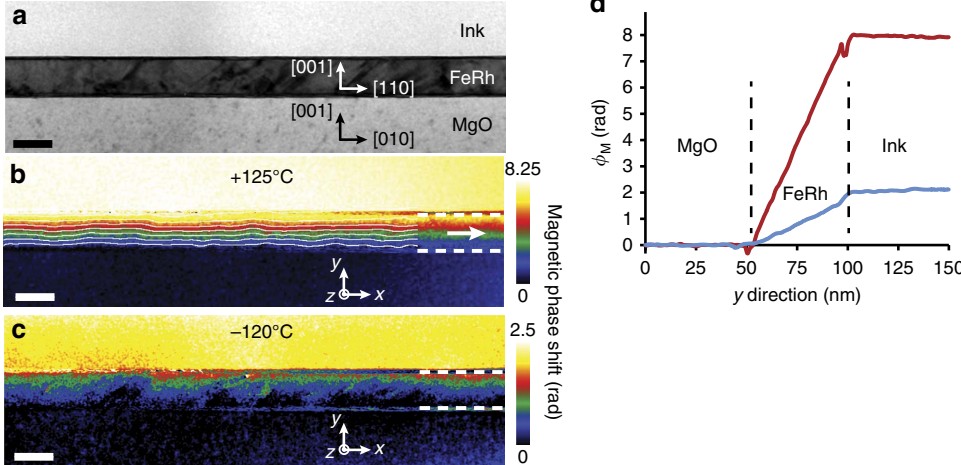

**Figure 2 | Electron holography images for the two extreme temperatures.** (**a**) Amplitude image of the studied area (630 nm × 190 nm). (**b**) Magnetic phase image obtained at +125 °C. The FeRh layer is between horizontal dashed lines at the right of the image. The white arrow and isophase lines parallel to the magnetic induction lines into the FeRh layer have been added for clarity. (**c**) Magnetic phase image obtained at −120 °C. (**d**) Magnetic phase shift profiles along the y direction averaged over the x direction for the whole field of view. The red and blue curves correspond to the profiles extracted at +125 °C and −120 °C respectively. Scale bars represent 50 nm.

sample temperature was then stabilized at +125 °C to get the FeRh layer into the FM state before applying an *in situ* magnetic field to saturate the specimen parallel to the [110] direction of FeRh (see Supplementary Note 3). A first set of 20 holograms was acquired during a temperature decrease down to −120 °C (cooling process) before recording a second set of 35 holograms during the heating process up to +125 °C. A third set of 22 holograms was recorded by decreasing the temperature down to −120 °C but after *in situ* magnetic saturation at +125 °C in the direction opposite to the previous one. All the holograms were acquired at the remnant state on the same area after temperature stabilization to perform a static study of the transition. The smallest temperature step between two successive holograms was 5 °C. We checked between the beginning and the end of the full EH experiment that the FeRh layer has not been modified by the irradiation of the electron beam by comparing the amplitude images, but also the magnetic phase images obtained in the same ferromagnetic state at +125 °C.

**Magnetic properties for the two extreme temperatures.** Figure 2 presents the amplitude and magnetic phase images for +125 and −120 °C. The field of view of the studied area is 630 nm × 190 nm. The contrast variations visible on the amplitude image (Fig. 2a) are commonly observed by TEM on FeRh layers and are attributed to the presence of defects as dislocations.

The ferromagnetic state clearly shows up on the magnetic phase image obtained at +125 °C with the appearance of a phase shift inside the FeRh layer (Fig. 2b). The magnetic induction lines displayed by the isophase contours parallel to the x direction demonstrate an induction aligned along the layer plane. As detailed in Supplementary Note 5, 3 nm thick layers of FeRh close to the top and bottom interfaces were affected by artefacts due to data processing and consequently were excluded from the analysis. The phase shift profile across the FeRh layer (that is, parallel to the y direction) averaged over the whole field of view in the x direction is given in Fig. 2d: the plateau in the ink and MgO areas indicate that there is no induction outside the film although the linear variation inside the layer proves that the induction is spread homogeneously within the FeRh film. This profile is the same for each position along the x axis and confirms that the magnetization is homogeneous all over the observed area.

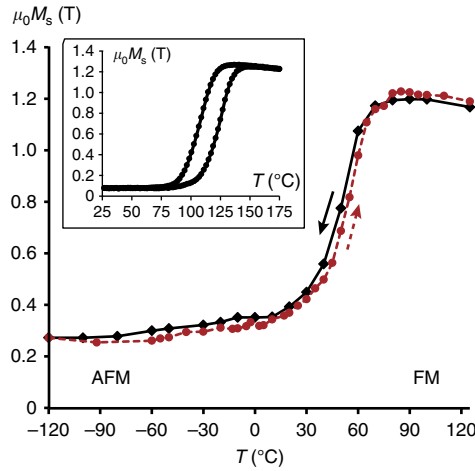

**Figure 3 | Local magnetization versus temperature loop.** x component of the magnetization as a function of temperature for a complete temperature cycle. The values are obtained at the remnant state from the slope of the magnetic phase shift averaged on the field of view detailed in Fig. 2. The cooling and heating series correspond to the black and red dashed lines respectively. In insert, the $M(T)$ loop measured by vibrating sample magnetometry on the whole FeRh sample from which the thin lamella has been extracted for electron holography experiments. This loop has been acquired with a 2 kOe field applied along the [110] axis of the FeRh layer.

According to equation (1), the integrated induction averaged over all the FeRh layer displayed in Fig. 2a can be obtained from the mean slope of the phase shift profile using a linear fit. The constant slope of $0.163 \pm 0.01$ rad nm$^{-1}$ in the FeRh layer corresponds therefore to an integrated induction of $107 \pm 6$ T nm distributed homogenously in the film thickness. The width $w$ of the TEM lamella being $90 \pm 5$ nm, the magnetization $\mu_0 M_x$ was calculated equal to $1.19 \pm 0.13$ T (equation (2)). This value is in very good agreement with previous magnetization values obtained at the FM state[10,12,27].

The magnetic phase image recorded at −120 °C (Fig. 2c) shows a strong decrease of the total phase shift. From its mean slope extracted on the whole FeRh layer, we obtained a residual

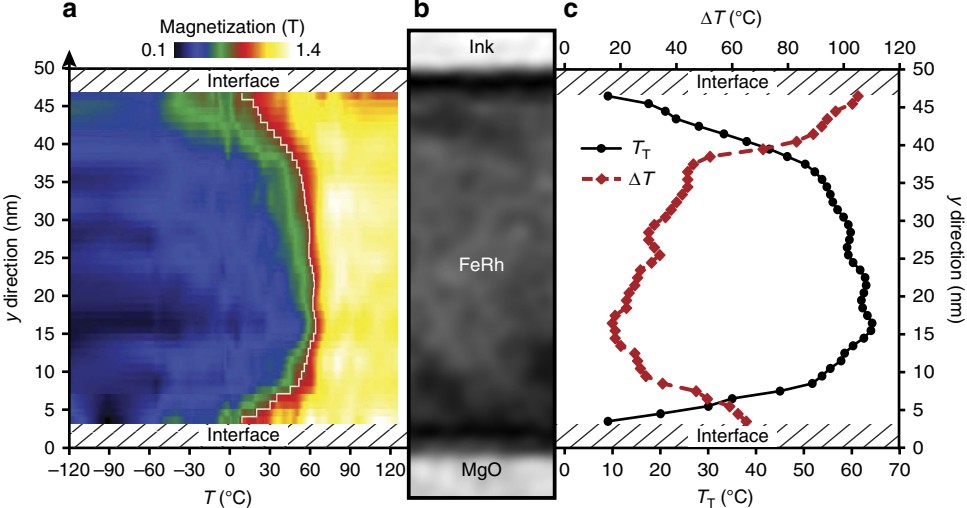

**Figure 4 | Evolution of the magnetization in the growth direction.** (**a**) Map of the magnetization as a function of the temperature and the position in the layer depth using the heating temperature series. The colour scale corresponds to the magnitude of the magnetization and the transition temperature is displayed by the white profile corresponding to magnetization of 0.75 T. (**b**) Amplitude image of the part of the FeRh layer used for the calculation. (**c**) Profiles of the transition temperature $T_T$ and the transition width $\Delta T$ as a function of the position within the FeRh layer.

magnetization parallel to the film equal to $0.28 \pm 0.13$ T. This value appears to be higher than previously reported[14,40].

**Local magnetization versus temperature loop.** In order to estimate the influence of the FIB preparation on the magnetic behaviour of the FeRh film, a magnetization versus temperature loop was acquired by EH and compared to VSM results. The analysis of the phase shift slope was therefore repeated on the same area of the FeRh layer displayed in Fig. 2 both when decreasing (cooling) and increasing (heating) the temperature. Figure 3 shows the complete loop $M_x = f(T)$ in the temperature range $-120\,^\circ\text{C} < T < 125\,^\circ\text{C}$. Error bars are not reported for better clarity of the plot and are in the range of $\pm 0.07$ and $\pm 0.1$ T at each point. While increasing the temperature, a plateau is observed around 0.28 T up to $-40\,^\circ\text{C}$. The magnetization increases slightly up to $20\,^\circ\text{C}$ then with a higher slope up to $45\,^\circ\text{C}$. An even steeper increase occurs up to $65\,^\circ\text{C}$ before slowing down up to a magnetization maximum just over 1.23 T at $80\,^\circ\text{C}$. Beyond $80\,^\circ\text{C}$, the magnetization follows a Curie–Weiss law usually observed in ferromagnetic materials. The trend is inversed when the temperature is decreased with a slight hysteresis of about $5\,^\circ\text{C}$ between the two curves. The width of the transition is about $60\,^\circ\text{C}$ for heating and cooling, indicating the coexistence of AFM and FM domains over a wide temperature range. By defining the transition temperature $T_T$ as the one corresponding to a magnetization of 0.75 T (intermediate value between AFM and FM states), $T_T$ is found equal to $55\,^\circ\text{C}$ ($\pm 5\,^\circ\text{C}$). The loop opening is measured at about $5\,^\circ\text{C}$ with an uncertainty of $\pm 5\,^\circ\text{C}$ if error bars are taken into account.

The inset in Fig. 3 presents the $M(T)$ loop measured by VSM on the whole FeRh layer from which the TEM lamella has been extracted. A 2 kOe magnetic field was applied during the VSM experiment along the same [110] direction as the one selected for induction measurements ($x$ direction). Important quantitative differences between the macroscopic and microscopic analyses are evidenced. As explained in the following, this result most probably originates from the difference in both investigated volume and sample geometry in the two techniques. Nevertheless, as in any TEM study, we cannot totally rule out a possible bias due to unexpected local inhomogeneities in the EH investigated

region. First, the transition rate, defined as $(M_{HT} - M_{LT})/M_{HT}$ where $M_{HT}$ is the high-temperature magnetization (FM state) and $M_{LT}$ is the residual magnetization at low temperature (AFM state), is 94% when measured macroscopically by VSM and $76 \pm 11\%$ when using the magnetization obtained by EH at $+125\,^\circ\text{C}$ and $-120\,^\circ\text{C}$. In addition, the transition temperature is about $108\,^\circ\text{C}$ in the macroscopic sample and about $55\,^\circ\text{C}$ in the thin lamella. Last, the cycle opening reaches $18\,^\circ\text{C}$ in the macroscopic film, while it is about $5\,^\circ\text{C}$ in the EH specimen.

As the macroscopic magnetization measured in the FM state by VSM is very similar to the one measured locally by EH ($M_{HT} = 1.27$ T and 1.19 T respectively), the difference in transition rate comes from the low temperature behaviour (that is, $M_{LT}$): a large FM component remains at low temperature in the TEM specimen used for EH experiments. Indeed, during the lamella preparation by FIB, the ion beam, even at low energy, creates two amorphous layers on each side of the thin lamella. The FM to AFM transition upon cooling, only present in the B2 phase[40], is impeded in these regions which remain in the FM state. Assuming a minimum thickness of 5 nm for each layer, these two damaged FM layers correspond to more than 10% of the total width of the thin lamella crossed by the electron beam (see Supplementary Fig. 5). The magnetization of the AFM state limited to the B2 phase thickness (80 nm) is therefore equal to 0.166 T giving a minimum conversion rate of $86 \pm 13\%$, that is, in reasonable agreement with the transition rate measured by VSM and others studies. This conversion rate would increase for thicker damaged layers with a magnetization of the AFM state approaching 0. The difference between the two transition temperatures measured by VSM and EH can also be attributed to the presence of the FIB damaged regions, which decrease the magnetic transition temperature due to their strong magnetic coupling. The small cycle opening is a mark of a quasi-reversibility of the transition process, which would require less energy. It most probably originates from the nanowire-like geometry (1D system) of the TEM lamella (section of $50 \times 90\,\text{nm}^2$ over several $\mu$m of length) than to the 2D film measured by VSM. The data recorded in EH thus suitably reproduce the magnetic behaviour of the film when finite-size effects and ion beam damaged regions are taken into account.

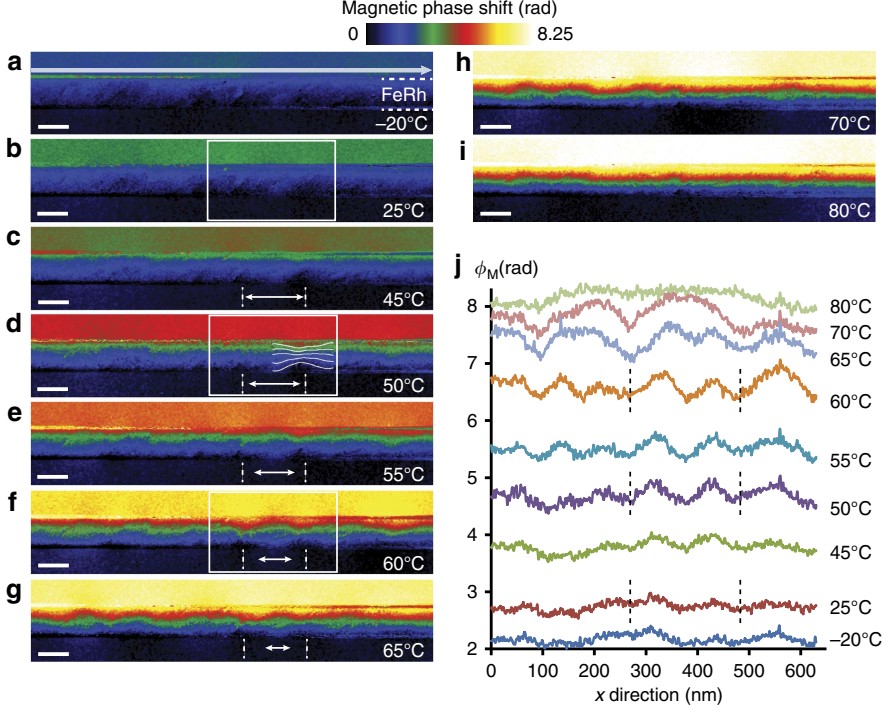

**Figure 5 | Main stages of the transition while increasing temperature.** (**a**–**i**) Magnetic phase images. The colour scale is common to all images and scale bars represent 50 nm. On the phase image recorded at 50 °C (**d**) are displayed the isophase lines parallel to the induction lines: the narrowing corresponds to a local increase of the phase variation and the appearance of ferromagnetic domains. The two dotted lines into the MgO part are centred on two ferromagnetic domains and the double arrow between them corresponds to the approximate width of the enclosed antiferromagnetic domain. The areas enclosed by the white rectangles for phase images at 25 °C (**b**), 50 °C (**d**) and 60 °C (**f**) are used for micromagnetic simulations presented Fig. 6. (**j**) Phase profiles extracted for different temperatures along the upper interface (large arrow on the magnetic phase image obtained at −20 °C).

**Evolution of the transition with FeRh layer depth.** EH allows for a deeper local analysis of the AFM/FM transition by extracting the magnetic information originating from smaller areas of the FeRh cross-sectional view (see Supplementary Fig. 3b–d). The evolution of the temperature transition along the growth direction has then been studied using a rectangular mask with dimension $x = 630$ nm and $y = 4$ nm elongated parallel to the interface with the MgO substrate. The area enclosed by the mask was shifted by 1 nm along the $y$ direction and, for each position, a complete loop $M_x = f(T)$ was extracted. Figure 4a presents the evolution of the AFM/FM transition from the bottom interface to the top surface using the heating temperature series (35 holograms from −120 to +125 °C). The first 3 nm close to both interfaces have been excluded because of artefacts in the data processing. The main result is the evidence of the heterogeneity of the transition: close to the interfaces, especially close to the top surface, the magnetic transition from the AFM state to the FM state starts at much lower temperature and is spread over a wider range of temperature than in the middle of the layer. The profile of $T_T$ along the growth direction has been extracted from Fig. 4a and is shown in Fig. 4c: $T_T$ is lower close to the interfaces than in the middle of the layer with a difference up to 50 °C. The core of the layer presents a 30 nm plateau with a mean value of $T_T$ slightly lower than 60 °C. $T_T$ decreases towards the interfaces down to 10 °C over a width of 5 to 10 nm. The transition width $\Delta T$ has been defined as the temperature range corresponding to an integrated induction between 0.5 and 1 T. By superimposing $\Delta T$ and $T_T$ profiles as a function of the position in the film, a clear match occurs: the lower $T_T$, the wider the transition. Similar results are obtained studying the cooling series.

## Discussion

Previous studies have shown for thin films that a FM state was stabilized and the transition temperature measured macroscopically evolved but the origin remained not entirely understood. If the presence of interfaces is considered to be at the origin of these experimental results[27,30,41], the stress effect[10,14,31] is also suspected to play a role. The properties of interfaces, in particular with capping layers, have been widely investigated in FeRh films, notably using XMCD-PEEM experiments. The occurrence of a persistent FM state in large surface areas near the surface, depending on the nature of the capping layer, was thus demonstrated[12,14,17,29]. In our case, the interdiffusion of the ink layer is most likely very limited and has no effect, as it was deposited at room temperature and the maximum temperature the system reached is 125 °C. Any structural defects have been seen in the FeRh layer at the top surface compared to the inner part. Similarly, a slight oxidation of the FeRh would spread over less than 2 nm and has not been observed. So even if we cannot exclude a slight effect of stress, symmetry breaking seems to be the most likely responsible for the magnetic behaviour changes. Regarding the MgO/FeRh interface, a diffusion of Mg might have occurred. In addition, the presence of structural defects such as misfit dislocations at this interface certainly promotes magnetic transition changes. These different mechanisms are sufficient to explain the slight asymmetry of the $T_T$ and $\Delta T$ profiles between both interfaces. We observed also that the interface effects extend up to 15 nm, that is, deeper into layer than previously assumed.

We thus demonstrate that both interfaces have a huge influence on the AFM/FM in the same way by reducing $T_T$ and broadening the temperature range necessary for the transition. This result is of large importance since it explains why various

experimental measurements observe a decrease of the transition temperature with the decrease of the layer thickness[27]. Our result thus predicts the difficulty to obtain an AFM/FM transition in FeRh nanoparticles of few nanometres diameter. In addition, interface effects have to be taken into account when FeRh layer is integrated in a device and coupled with adjacent (magnetic) layer[20,22,23].

The magnetic phase images corresponding to the main stages of the transition from low temperatures to high temperatures are presented (Fig. 5). The studied area is identical to the one analysed in Fig. 2 and the colour scale is common to all images. The gradual increase of the total phase shift with increasing temperature reflects the appearance of the FM state as discussed previously. At $-20$ and $+25\,°C$, the phase images are relatively similar to the one obtained at $-120\,°C$ with only slightly larger phase value due to the beginning of the AFM/FM transition at surfaces or interfaces. The magnetic configuration varies more significantly from $45\,°C$. Interfaces with an almost complete FM state exhibit a higher phase variation (that is, magnetization) than in the core of the film. In addition, inhomogeneities of the phase shift appear in direction parallel to the interfaces ($x$ direction): some areas with narrow induction lines (that is, corresponding to larger phase variations) become visible within the layer of FeRh (see isophase lines on the phase image recorded at $50\,°C$). They correspond to the nucleation of FM areas. These local FM regions are in addition coupled with other small variations outside the film near the interfaces corresponding to their leak field. In Fig. 5, two FM domains that remain for temperatures between 45 and $65\,°C$ are identified by dotted lines (drawn in the MgO part for better clarity). The distance between the cores of these FM domains enclosed within an AFM matrix is about 100 nm. The double white arrow indicates the approximate width of the AFM area between the two FM domains. This width decreases gradually implying a lateral extension of the FM domains before a sudden disappearance of the remaining AFM area between 65 and $70\,°C$, the coalescence of the FM areas being favoured due to the magnetic coupling between them that promotes the transition from AFM to FM state.

This AFM/FM transition mechanism though FM domain nucleation within the FeRh layer was confirmed by the study of leak fields spreading out of the FM domains. They are evidenced by extracting the magnetic phase profile along the upper interface (white arrow on the phase image at $-20\,°C$). The profiles obtained at different temperatures are given in Fig. 5j and show the appearance of oscillations whose maxima and minima are centred on the FM and AFM areas respectively. Oscillations appear clearly from $35\,°C$ (not shown), corresponding to the end of nucleation process, increase in amplitude (FM domain growth) and begin to disappear from $65\,°C$ (FM domains coalescence) and completely at $80\,°C$. The most remarkable result is the constant spatial periodicity (of about 100 nm) of the FM domains reflecting the regular repetition of the FM and AFM areas. Note that the value of this period is comparable to the distance between misfit dislocations allowing a complete relaxation of a FeRh layer on MgO. We may assume that the nucleation of these FM domains occur on these structural defects. Our results confirm that the FM phase nucleates at the interfaces and propagates into the bulk as the temperature rises[31]. However we demonstrate the existence of a second mechanism corresponding to the nucleation of small FM domains into the FeRh layer. This mechanism cannot be observed if using probe-based methods.

Micromagnetic simulations with the OOMMF package (http://math.nist.gov/oommf) were used to confirm the appearance of the FM domains in the AFM matrix. These simulations are performed at 0 K, the temperature being introduced through a magnetic model based on by EH data at a given temperature. In Fig. 6 we compare the experimental magnetic phase shift at 25, 50 and $60\,°C$ with the ones calculated from three magnetic configurations. Experimental data were taken from the area marked by a white rectangle in Fig. 5b,d,f. The presence of the disordered FM layers was taken into account (not shown in the 3D scheme for clarity), as well as the effect of the interfaces on the transition. The distance between the FM domains in the micromagnetic calculations is 100 nm and their lateral size was adjusted accordingly to the experimental phase images (20 nm at $50\,°C$, 40 nm at $60\,°C$). All other parameters of the simulation are given in Supplementary Note 6. A very good quantitative agreement between the simulated and experimental phase images is obtained for each temperature. No FM domains are observed at $25\,°C$. At $50\,°C$, the FM domains have nucleated within the layer while thickness and magnetization of the ferromagnetic interfaces have increased. At $60\,°C$, these interfacial areas have completed the transition to the FM state while the FM domains within the layer extend laterally. The small differences occurring between the simulated and experimental phase images is the result of our hypothesis regarding the thickness of the interfacial zones and also may be due to the shape of the FM domains that surely are not perfectly rectangular as proposed.

Successive schemes of the AFM/FM transition have therefore been determined by EH and observed in the $M(T)$ loop shown in Fig. 3: the induction starts to increase with the interface transition ($-40\,°C < T < 20\,°C$), it accelerates slightly with nucleation of periodic FM domains from $25\,°C$ even if the transition at interfaces is not achieved, then an abrupt change from $35\,°C$ corresponding to the FM domain growth within the AF matrix and a slower increase from $65\,°C$ during the coalescence phase until the complete disappearance of the AFM state. The observations during cooling process show a reverse transition involving the same mechanisms but with a slight shift towards the low temperatures of about $5\,°C$.

Thus the cross-sectional observation of an FeRh layer at the nanoscale using in situ EH upon heating and cooling cycles allows us to obtain a clear and quantitative description of the AFM/FM transition process. The analysis of magnetic phase images evidenced an evolution of the transition temperature from the bottom interface with the substrate to the top surface associated to a regular alternation of FM domains in an AF matrix. Our results bring insights for in operando analysis of magnetic devices. Beyond these findings, the paper describes a very promising approach to investigate the mechanisms of phase transitions in various magnetic systems such as MnAs[42] at all pertinent scales.

## Methods

**Sample growth and thin lamella preparation.** The 50 nm thick FeRh layer was grown on an MgO (001) substrate by DC sputtering using a co-deposition process from two pure Fe and Rh targets. The film was deposited at $550\,°C$ and in situ annealed for 6 h at $800\,°C$. For EH experiments, a cross-sectional lamella was prepared by FIB to ensure a uniform thickness crossed by the electron beam. The FeRh was protected by a 150 nm thick ink layer and a 250 nm thick Pt layer to avoid damages and charge accumulation during the thinning process. The lamella was then extracted and thinned down to about 100 nm to get electron transparency with a final step at low energy to minimize irradiation damages and amorphization of the surfaces.

**Magnetometry.** The AFM/FM transition was checked by VSM of a quantum Design Physical Properties Measurement System (PPMS) on the whole sample elaborated by sputtering. A continuous 2 kOe magnetic field was applied along the in plane [110] direction of FeRh during all magnetization measurements as a function of temperature.

**TEM and heating-cooling sample holder.** The EH experiment was performed using a Hitachi HF 3300C microscope operating at 300 kV. Dedicated Lorentz modes combined with the B-core corrector allow achieving a 0.5 nm spatial

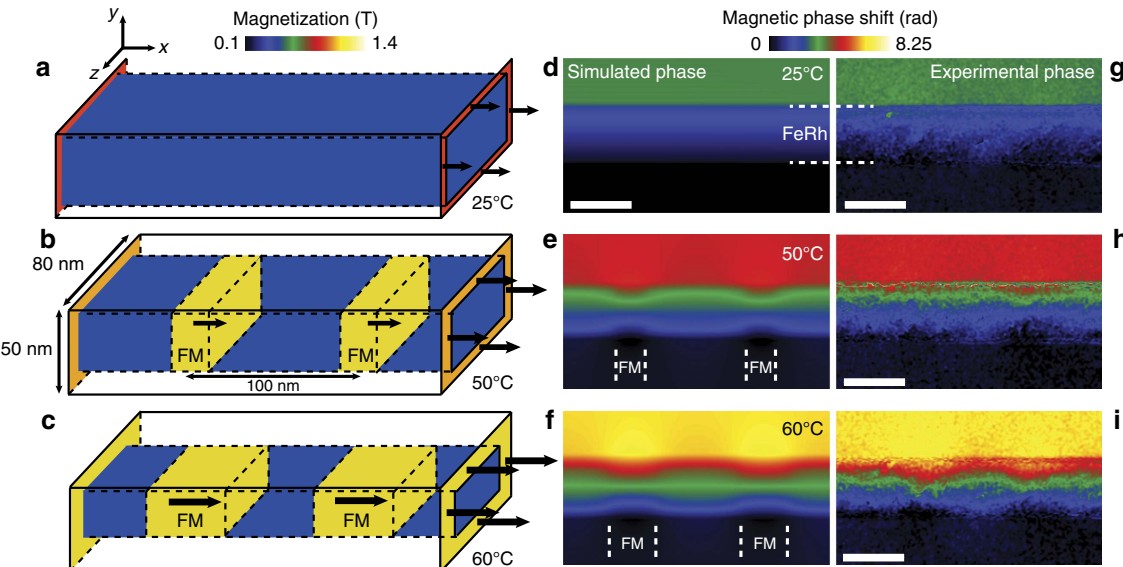

**Figure 6 | Micromagnetic simulations and comparison with experimental phase images.** (**a–c**) 3D schemes used for the micromagnetic simulations of the area enclosed by the white rectangle in Fig. 5b,d,f (25, 50 and 60 °C respectively). The ferromagnetic damaged surfaces are not represented for clarity. (**d–i**) Comparison between simulated magnetic phase images calculated from micromagnetic simulations and (**h–j**) experimental magnetic phase images obtained at same temperatures. Scale bars represent 50 nm.

resolution in a field-free environment (less than 10 Oe). All the holograms were recorded in a 2 biprism configuration[43] with a fringe spacing set to 0.9 nm. The sample holder is a single tilt Gatan HC3500 holder that permits temperature control by PID (Proportional Integral Derivative) from − 150 to + 250 °C.

**Data treatment.** Phase and amplitude images were extracted from the holograms using homemade software based on Fourier analysis. The size of the digital mask used in the FFT was chosen to obtain a spatial resolution of 2.5 nm on phase and amplitude images. Amplitude images, which remain the same regardless of the applied temperature, were automatically realigned by a cross-correlation method. This realignment was then applied on the phase images. The electrostatic $\phi_E$ and magnetic $\phi_M$ contributions from the total phase shift were separated using the phase images obtained at 125 °C after having saturated the magnetization of the layer in opposite directions: $\phi_E$ remains the same while $\phi_M$ sign changes. The half sum of these images provides $\phi_E$, which is then subtracted from all the phase images: only $\phi_M$ contribution is then retained.

The full procedure for induction quantification from magnetic phase images is detailed in Supplementary Note 5.

**Micromagnetic simulations.** Micromagnetic simulations were performed using the OOMMF 3D package (http://math.nist.gov/oommf). The universe used for calculation has dimensions of 660 nm along the x direction, 330 nm along the y direction and 330 nm in the z direction (direction parallel to the electron beam) with a size cell of $2 \times 2 \times 2$ nm. The simulated FeRh layer has a thickness of 50 nm along the growth direction (y direction) and a width (thickness crossed by the electron beam) of 90 nm from estimations during the FIB process. This width is divided into three layers: a layer of 80 nm thick composed of ordered FeRh presenting the AFM/FM transition separates two others layers of 5 nm thick perpendicular to the electron beam corresponding to the amorphized surface of the lamella by the FIB preparation. Three temperatures (25, 50 and 60 °C) have been investigated. The damaged layers were kept in a ferromagnetic state for all applied temperature with a magnetization equal to 1.19 T. This magnetization is oriented along the x direction corresponding to the direction of initial saturation. The volume of ordered FeRh where the magnetization evolves with the temperature is in the form of a rectangular wire with a $50 \times 80$ nm$^2$ section and a core-shell structure. The shell takes into account the fact that the magnetic transition at surfaces/interfaces is different from that in the core in which the FM and AFM areas nucleate and coalesce. The shell thickness was taken as the average of the measured widths as a function of temperature (Fig. 4a), and considered to be uniform over the entire section of the layer in the simulations for simplicity. The thickness of the shell and its magnetization parallel to the x direction increase gradually with temperature (6 nm/0.8 T, 9 nm/1 T and 12 nm/1.19 T for 25, 50 and 60 °C respectively). The mean magnetization of the FM domains (0.94 T at 50 °C and 1.07 T at 60 °C) was extracted from the central portion of Fig. 4a and set parallel to the x direction, while the magnetization of AFM areas is considered equal to 0. The period of the FM domains is 100 nm and their lateral size was adjusted accordingly to the experimental phase images (20 nm to 50 °C, 40 nm to

60 °C). From the micromagnetic simulations of this system, the components of the magnetization and the dipolar field have been extracted to calculate the total magnetic induction. After integrating the components perpendicular to the electron beam along the electron path, the corresponding simulated phase image was calculated using Aharonov–Bohm equations and compared directly to experimental magnetic phase images.

**Data availability.** All relevant data are available from the authors.

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

## Acknowledgements

The authors acknowledge the French National Research Agency under the 'Investissement d'Avenir' programme reference No. ANR-10-EQPX-38-01, the 'Conseil Regional Midi-Pyrénées', the European FEDER for financial support within the CPER programme, and the European Union under the Seventh Framework Program under a contract for an Integrated Infrastructure Initiative Reference 312483-ESTEEM2. This work was supported by the French national project EMMA (ANR12 BS10 013 01) and the French microscopy network METSA.

## Author contributions

C.G. and M.J.C. conceived the research, M.C. deposited epitaxial FeRh thin films and performed VSM measurement, C.G., L.A.R. and D.R. performed the electron holography experiments, C.G. performed the data treatment and analysis, C.G. and N.B. performed micromagnetic simulations, C.G., M.J.C. and B.W.-F. discussed and wrote the manuscript.

## Additional information

**Competing interests:** The authors declare no competing financial interests.

**Publisher's note**: 

