## [Peer Review File · Nature Communications]

Reviewers' comments:

Reviewer #1 (Remarks to the Author):

In this report the authors present a study of the spatially resolved magnetic structure in a thin film of the binary alloy FeRh. Using electron holography (EH) they observe an inhomogeneous magnetic ordering along the growth direction. Furthermore, the in-plane domain structure is also resolved. A micromagnetic model is constructed which describes features of the experimental data.

The paper can be divided into two sections: a description of the electron holography measurement and a study of the FeRh system.

The discussion of the EH technique is well described and is certainly of interest to those working in the magnetic thin film community. EH is a relatively underexploited technique in the thin film magnetism community and so the accessible work presented here is most timely and welcome.

Turning to the discussion of the FeRh system the relevance of the work is less clear as presented. The properties of FeRh are well known to be sensitive to a wide range of variables: crystal structure, disorder, doping, applied field etc. the behavior of the system, as evidenced by the EH is very different to those obtained by VSM and reported in this work. Transition temperature and temperature hysteresis are very different. Commendably, the authors attempt to address these discrepancies in the manuscript but this falls short of convincing this referee that the FeRh being investigated is representative of FeRh which has not undergone the detailed processing required by the EH technique. Figure 3 would appear to support this point: the preparation for the EH measurements is clearly highly disruptive.

Specific comments and queries:

The observation of a (001) reflection is not sufficient to describe the quality of the B2 ordered phase. Much work in the literature describes how the chemical order parameter strongly affects the properties of the film. Given the processing of the film this is likely to be significant and uncharacterized in the present study.

How important is it that all of the EH measurements were taken at the remanent state? This is pertinent in this magnetic field is known to influence the magnetic transition.

The manuscript describes a residual magnetization in the sample at low temperature throughout the film. This appears to contradict the literature which suggests that this component is localized at the interfaces?

In Fig.4 the authors spatially resolve the transition temperature within the film. Recent studies on FeRh have introduced doping gradients into the film [1] to create a moveable phase boundary to generate results similar to those presented here. Can techniques such as XPS be used to map out the elemental composition resolved in the film? Again it suggests the film is significantly damaged.

The manuscript suggests caution is required when interpreting XMCD-PEEM data given its surface sensitivity. My understanding is that such measurements, on high quality films, give domain structures comparable in size to those observed in the 'good' section of the EH film suggesting that they are reliable. Indeed XMCD-PEEM studies have gone on to show the influence of surface regions (capping layers Ref#29). The authors should consider a fuller discussion of this extensive body of literature.

The micromagnetics discussion should make it more clear that the modelling is a zero temperature calculation in which the temperature is introduced by constructing a magnetic model suggested by the EH data at a given temperature.

Fig1. The yellow and white colors do not provide good contrast in the figure.

In summary, whilst I commend the authors work, I am not convinced that the sample investigated is relevant to research on FeRh or magnonics as is alluded to without serious justification.

1. Le Graët, C. et al. Temperature controlled motion of an antiferromagnet- ferromagnet interface within a dopant-graded FeRh epilayer. APL Mater. 3, 41802 (2015).

Reviewer #2 (Remarks to the Author):

The data shown here from electron tomography (EH) are undeniably the most detailed maps of the magnetic moment in metamagnetic FeRh films shown to date. I feel that both the data itself, and relevant conclusions about the growth direction domain dynamics, are significant to the scientific community - however some technical points must be address before continuing further.

- For those not familiar with EH there should be included an image/diagram/schematic that shows the sample (lamella) that was used for measurement. This depiction should also include designation of the region on the sample that was measured for clarity.

- Some statements in the text are inaccurate and should be revisited..

Specifically in paragraph #2 - "However, no direct observation was performed at the nanoscale to investigate not only the layer core but also the transition distribution along the growth direction: all previous experiments deduce from surface observations or macroscopic measurements a global behaviour of the transition inside the layer." Experiments using polarized neutron reflectometry have been able to map the magnetic moment along the growth direction with sub-nm resolution.

I think it important to discuss the recent works by S.P. Bennett et. Al. which map with high detail the depth dependent magnetization of FeRh films in Scientific Reports ("Giant Controllable Magnetization Changes Induced by Structural Phase Transitions in a Metamagnetic Artificial Multiferroic" Sci. Rep. 6, 22708 (2016) & "Direct evidence of anomalous interfacial magnetization in metamagnetic Pd doped FeRh thin films" Sci. Rep. 5, 9142 (2015) - I also think it is very important to make clear that neutrons average over the whole sample and small local variations in the transition interfaces are therefore imperceivable. However these EH images show clearly how the transition is evolving microscopically at these interfaces - allowing the detailed analysis of the domain structure.

- This data also seems to refute claims that a topological/surface effect could cause surface terminated magnetization in FeRh. Could this be elaborated to make that conclusion as well?

- Elaborate more on the induction effects used to describe the following conclusion on line 221 "Therefore interfaces not only lower the transition temperature, but also make this transition more difficult to be completed" - Describe in more detail how the interface is making the transition harder to complete?

- The use of letters to denote regions of interest in the phase shift diagram in fig. 3 is confusing. I recommend using temperature ranges and removing the letters.

- As a general note, for clarity to the community, magnetic moments should be delineated within the article as emu/cc. I also feel it would be helpful to convert to volumetric magnetization units to compare to VSM data.

Reviewer #3 (Remarks to the Author):

The manuscript authored by Dr. Gatel and collaborators provides some useful information for understanding of the magnetic phase transition in an FeRh film. The electron microscopy observations definitely show a variation of phase transition temperatures within the FeRh layer sandwiched by nonmagnetic substances. However, my opinion is that this study is still incomplete, because of significant technical concerns that need further examinations. Although the discussion using the magnetic induction maps provide beneficial information for the study of this particular system, I am nevertheless not sure whether this manuscript offers wide interests to readers of nature communications. This paper needs significant revisions to address the technical concerns and show the suitability for publishing in nature communications, otherwise should be published in a specialized journal of materials science without further delays.

1) page 2: Although the sample thickness was roughly estimated at 90 nm, which was deduced from SEM images such as shown in Fig. S1, the sample prepared by using focused ion beam must be wedge-shaped: it is accordingly difficult to evaluate the true thickness in the portion of FeRh using the SEM images. When the sample thickness is unclear, it is difficult to discuss about the magnetic induction values.

2) page 3: Electron holography must be sensitive to magnetic induction B_x , rather than magnetization M_x . The relationship of eq.2 is invalid unless the effect of leakage magnetic field, which must be present outside of the sample, is incorporated correctly.

3) page 3: "The contrast layers...are attributed to the presence of defects (dislocations...)". As the authors indicate, structural imperfections appear to influence the magnetic phase transition significantly. The "presence of defects" should be discussed in greater detail.

4) page 3: My naïve question is, since the layer of MgO is of insulator, it must be strongly electric-charged by electron illumination. However, Fig. 2d does not show an appreciable phase shift due to electric charging in the MgO layer. The result is not consistent with the insulating nature in the MgO layer. Provided that the phase shift due to electric charging was already removed from the curves shown in Fig. 2d, the slope in the FeRh area must be affected accordingly.

5) page 4: It is unclear why the "antiferromagnetic" state shows a definite magnetization 0.28 T. Is this state "ferrimagnetic"?

6) page 5: The authors ascribe the disagreements in the shape of M-T curves, shown in Fig.3, to the structural damage that may be induced by focused ion beam. To justify this picture, both the magnetization and depth of the damage layer should be determined accurately. The effect of structural disorder on the Neel/Curie temperatures in this compound should be mentioned more deeply, in order to support the authors' conclusion. In addition, we would like more reasonable grounds as to why the temperature hysteresis and the transition point are reduced significantly in the foil sample.

7) page 7: What kind of "symmetry breaking" occurs in the FeRh sample.

8) page 7: "They correspond to the nucleation of FM areas." I guess, a significant volume change (1%) associated with the ferromagnetic/antiferromagnetic transition provides undesired diffraction contrasts, and accordingly additional phase shift in electron holography data. This point should be addressed.

For the reviewers

We first want to thank the reviewers for their careful reading of our manuscript and their useful comments, including positive appreciations and criticisms. We detail hereafter our answer to their different requests and how we changed the manuscript accordingly.

Answer to Reviewer #1 :

General comment: The reviewer points to the difficulty of comparing macroscopic and local properties. Indeed, the VSM results showing the magnetization versus temperature plot averaged in a 2mmx3mm area show differences with the results by EH measurements collected on a much smaller area (50nm x 90nm). However, the presented results are rather complementary, owing to the different scales of observation, than opposite. We agree with the reviewer that TEM related techniques are not the most suitable for collecting statistical information and are likely to emphasize micro/nanoscale inhomogeneities. However, as surprising as the observed differences may seem, they mainly concern the decrease of the transition temperature and the narrowing of the hysteresis cycle for the FIB prepared specimen.

In order to take the reviewer comment into account, we add after *“Important quantitative differences between the macroscopic and microscopic analyses are evidenced.*

the following text:

“As explained in the following, this result most probably originates from the difference in both investigated volume and sample geometry in the two techniques. Nevertheless, as in any TEM study, we cannot totally rule out a possible bias due to local inhomogeneities in the EH investigated region.”

Indeed, as the in-situ heating and cooling EH experiments together with the required processing are very long procedures, (200 images to collect, adjust and process) examination of many different regions is hardly an option.

The preparation of TEM specimen by FIB (used for our EH measurements) may again seem “highly disruptive” to the reviewer but the process is in fact now very well controlled and so is the thickness of the damaged layer. We can just argue here that important and reliable results, even in materials very sensitive to defects as semiconductors, obtained from FIB specimens prepared according to the same procedure, are widely reported in the literature and the results are commonly analyzed by taking into account the presence of the damaged layers on both sides of the specimen. In this article, we have considered 5nm thick damaged layers on both sides of the crystalline part and this explains the magnetic behaviour of the layer (discrepancy between VSM and EH cycles for example). The resulting magnetic coupling acts as a strong external magnetic field which decreases the transition temperature as shown using VSM in many publications.

Moreover, many articles have been published, in Nature Comm and elsewhere,

especially on in-situ TEM experiments with lamella prepared by FIB.

Debora Keller et al., "Assessment of off-Axis and in-Line Electron Holography for Measurement of Potential Variations in Cu(In,Ga)Se₂ Thin-Film Solar Cells," *Advanced Structural and Chemical Imaging* 2, no. 1 (January 13, 2016): 1, doi:10.1186/s40679-015-0015-5.

Tim Grieb et al., "Determination of the Chemical Composition of GaNAs Using STEM HAADF Imaging and STEM Strain State Analysis," *Ultramicroscopy* 117, no. 0 (2012): 15–23, doi:10.1016/j.ultramic.2012.03.014.

Martin Hýtch et al., "Nanoscale Holographic Interferometry for Strain Measurements in Electronic Devices," *Nature* 453, no. 7198 (June 19, 2008): 1086–89, doi:10.1038/nature07049.

Jonathan D. Poplawsky et al., "Structural and Compositional Dependence of the CdTeSe_{1-x} Alloy Layer Photoactivity in CdTe-Based Solar Cells," *Nature Communications* 7 (July 27, 2016): 12537, doi:10.1038/ncomms12537.

Maria Koifman Khristosov et al., "Sponge-like Nanoporous Single Crystals of Gold," *Nature Communications* 6 (November 10, 2015): 8841, doi:10.1038/ncomms9841.

Myung-Geun Han et al., "Interface-Induced Nonswitchable Domains in Ferroelectric Thin Films," *Nature Communications* 5 (August 18, 2014): 4693, doi:10.1038/ncomms5693.

Specific comments and queries:

- The observation of a (001) reflection is not sufficient to describe the quality of the B2 ordered phase. Much work in the literature describes how the chemical order parameter strongly affects the properties of the film. Given the processing of the film this is likely to be significant and uncharacterized in the present study.

The referee is right; we do not describe the quality of the chemical order in figure 1. We just mention that the 001 reflection reveals the presence of the B2 phase. This phase is present in a large range of composition in the FeRh phase diagram. However, it is just in a very small range of composition (around 48 to 56 % Rh) that the B2 crystal phase is in a AFM magnetic state. So the B2 phase is a necessary but not sufficient condition. We analyzed our specimens in X-ray diffraction to check the presence of a single phase but we did not systematically quantify the degree of order as the presence of the AFM phase, measured by VSM measurements and EH, is strongly related to a high degree of order.

- How important is it that all of the EH measurements were taken at the remnant state? This is pertinent in this magnetic field is known to influence the magnetic transition.

The sample has been magnetically saturated at the beginning of the experiment and the shape anisotropy of the thin lamella guaranties the stability of the magnetic state. The remnant state is therefore equivalent to the saturation state at high temperatures. As a consequence, the magnetic transition was not influenced by a magnetic field in EH.

- The manuscript describes a residual magnetization in the sample at low temperature throughout the film. This appears to contradict the literature which suggests that this component is localized at the interfaces?

We did observe a strong effect (considerable decrease of the transition temperature) at interfaces, as shown in figure 4. Note that this figure has now been modified to avoid misunderstanding (see answer below). Concerning the persistence of a residual magnetization in the core of the film, it is mainly attributed to the presence of the FIB damaged layers (parallel to the cross sectional plane and so to the image plane), already mentioned in our answer to previous comments. As written in the manuscript: *“Assuming a minimum thickness of 5nm for each layer, these two damaged FM layers correspond to more than 10% of the total width of the thin lamella crossed by the electron beam (see scheme in the micromagnetic part of SI). The magnetization of the AFM state limited to the B2 phase thickness (80nm) is therefore equal to 0.166 T giving a minimum conversion rate of 86±13%, i.e. in reasonable agreement with the transition rate measured by VSM and others studies.”*

- In Fig.4 the authors spatially resolve the transition temperature within the film. Recent studies on FeRh have introduced doping gradients into the film[1] to create a moveable phase boundary to generate results similar to those presented here. Can techniques such as XPS be used to map out the elemental composition resolved in the film? Again it suggests the film is significantly damaged.

We believe that figure 4 is misleading and we changed it. This figure shows the profile of the transition temperature along the film growth direction. It emphasizes the lowering of this temperature close to the top and bottom interfaces. In our case, no capping layer nor dopant were used, only a protection coating layer (ink) at the surface of the film, which is not expected to diffuse inside the film, and of course there is also a possible diffusion of Mg and Fe atoms at the bottom interface. This can be measured using energy dispersive X-ray spectroscopy (EDX) for instance (XPS would be more favorable for the near surface region). Different authors have for instance observed Rh enrichment at the bottom interface due to diffusion of Fe in MgO.

- The manuscript suggests caution is required when interpreting XMCD-PEEM data given its surface sensitivity. My understanding is that such measurements, on high quality films, give domain structures comparable in size to those observed in the ‘good’ section of the EH film suggesting that they are reliable. Indeed XMCD-PEEM studies have gone on to show the influence of surface regions (capping layers Ref#29). The authors should consider a fuller discussion of this extensive body of literature.

We agree that this sentence may seem inappropriate taking into account the numerous results (cited in ref 12,14, 17, 29 for instance) obtained on the properties of near surface regions in FeRh films examined by XMCD-PEEM. Our purpose here was much more general than the case of FeRh. We suppressed the following sentence to avoid misunderstanding *“It also highlights the caution with which the surface technics (XMCD-PEEM,...) results must be analyzed as these techniques probe areas close to the surface thus influenced by the interface effects. The measured properties may then not correspond properly to the ones of the inner layer.”*

and replaced in the previous paragraph, the sentence:

“A capping layer also modifies the transition temperature but in different ways depending on its nature....”

by

“The properties of interfaces, in particular with capping layers, have been widely

investigated in FeRh films, notably using XMCD-PEEM experiments. The occurrence of a persistent FM state in large surface areas near the surface, depending on the nature of the capping layer, was thus demonstrated. (12,14, 17, 29)”

- The micromagnetics discussion should make it more clear that the modelling is a zero temperature calculation in which the temperature is introduced by constructing a magnetic model suggested by the EH data at a given temperature.

We agree with the referee that the OOMMF micromagnetic software does not take into account the temperature effects and that the evolution of the micromagnetic state in the FeRh layer is added by changing the ratio between the F/AF layers. In order to make it clearer we added after « *Micromagnetic simulations with the OOMMF package⁴⁰ were used to confirm the appearance of the FM domains in the AFM matrix.* »

the following text:

“These simulations are performed at OK, the temperature being introduced through a magnetic model based on by EH data at a given temperature. »

- Fig1. The yellow and white colors do not provide good contrast in the figure.

The colors have been changed from yellow to orange to provide enhanced contrast.

- In summary, whilst I commend the authors work, I am not convinced that the sample investigated is relevant to research on FeRh or magnonics as is alluded to without serious justification.

We removed the reference to magnonics in our manuscript as the FeRh sample indeed seems far from this application:

“This periodic FM/AFM magnetic configuration may find interesting applications in magnonic devices which require a periodic modulation of magnetic properties at nanoscale.”

“and the potential use of FeRh layer as magnonic crystal”

Reviewer #2 (Remarks to the Author):

The data shown here from electron tomography (EH) are undeniably the most detailed maps of the magnetic moment in metamagnetic FeRh films shown to date. I feel that both the data itself, and relevant conclusions about the growth direction domain dynamics, are significant to the scientific community - however some technical points must be address before continuing further.

- For those not familiar with EH there should be included an image/diagram/schematic that shows the sample (lamella) that was used for measurement. This depiction should also include designation of the region on the sample that was measured for clarity.

A scheme has been added in the SI (figure S2).

- Some statements in the text are inaccurate and should be revisited. Specifically in paragraph #2 - "However, no direct observation was performed at the nanoscale to investigate not only the layer core but also the transition distribution along the growth direction: all previous experiments deduce from surface observations or macroscopic measurements a global behaviour of the transition inside the layer ». Experiments using polarized neutron reflectometry have been able to map the magnetic moment along the growth direction with sub-nm resolution.
- I think it important to discuss the recent works by S.P. Bennett et. Al. which map

with high detail the depth dependent magnetization of FeRh films in Scientific Reports ("Giant Controllable Magnetization Changes Induced by Structural Phase Transitions in a Metamagnetic Artificial Multiferroic" Sci. Rep. 6, 22708 (2016) & "Direct evidence of anomalous interfacial magnetization in metamagnetic Pd doped FeRh thin films" Sci. Rep. 5, 9142 (2015). I also think it is very important to make clear that neutrons average over the whole sample and small local variations in the transition interfaces are therefore imperceivable. However these EH images show clearly how the transition is evolving microscopically at these interfaces - allowing the detailed analysis of the domain structure.

By "Direct observation", we meant imaging techniques mainly – but we agree with the reviewer that this can be misinterpreted. We changed the sentence

"However, no direct observation was performed at the nanoscale to investigate not only the layer core but also the transition distribution along the growth direction: all previous experiments deduce from surface observations or macroscopic measurements a global behaviour of the transition inside the layer ».

by

"In addition, the magnetic profile along the growth direction of FeRh films, for both undoped and doped films, was analyzed by polarized neutron reflectometry (PNR) with a sub-nm resolution.^{12,29,32,33} Spectra were suitably fitted by taking into account both a top and a bottom interface regions (around 8 nm thick) surrounding the core of the film."

The next sentence was also slightly modified :

"In contrast with the previous experimental methods, electron holography (EH) is able to provide quantitative imaging on the local magnetization, thanks to its high sensitivity to the magnetic signal³²⁻³⁵. Besides, information can be extracted across the entire film thickness with unrivalled spatial resolution notably along the interfaces compared to PNR."

- This data also seems to refute claims that a topological/surface effect could cause surface terminated magnetization in FeRh. Could this be elaborated to make that conclusion as well?

The reviewer probably refers to surface termination effect that has been predicted and experimentally observed at the very surface of thin films. In our case, the hologram regions corresponding to the 3nm thick areas close to the interfaces cannot be processed without introducing artefacts but we believe that the surface/volume ratio is too weak here to be concerned by the topological effect as mentioned in "Federico Pressacco et al., "Stable Room-Temperature Ferromagnetic Phase at the FeRh(100) Surface," *Scientific Reports* 6 (March 3, 2016): 22383, doi:10.1038/srep22383 » for instance "

- Elaborate more on the induction effects used to describe the following conclusion on line 221 "Therefore interfaces not only lower the transition temperature, but also make this transition more difficult to be completed" - Describe in more detail how the interface is making the transition harder to complete?

We have removed the following misplaced sentences as the discussion was detailed further in the text.

«Therefore interfaces not only lower the transition temperature, but also make this transition more difficult to be completed. We can assume that these effects are even stronger in the non-analysed areas at the interfaces and lead to a disappearance of the transition accompanied of a stabilization of the FM state.^{12,14,27,29,39} »

The discussion on the influence of interfaces is presented in the text below figure 4 and its different origins are introduced. Owing to the results presented in figures 5 and 6, we also considered the possible influence of misfit dislocations.

- The use of letters to denote regions of interest in the phase shift diagram in fig. 3 is confusing. I recommend using temperature ranges and removing the letters.

Figure 3 has been modified and the text has been changed to remove all the references to the regions of interest.

- As a general note, for clarity to the community, magnetic moments should be delineated within the article as emu/cc. I also feel it would be helpful to convert to volumetric magnetization units to compare to VSM data.

We think that using the SI units is more appropriate in our case as Tesla is the unit used in the Aharonov-Bohm equation, which describes the link between the induction and the phase shift. We have modified the following sentence: « *The width w of the TEM lamella being 90 ± 5 nm, the magnetization $\mu_0 M_x$ was calculated equal to 1.19 ± 0.13 T* »

by

« *The width w of the TEM lamella being 90 ± 5 nm, the magnetization $\mu_0 M_x$ was calculated equal to 1.19 ± 0.13 T (955 emu/cm³)* »

Reviewer #3 (Remarks to the Author):

The manuscript authored by Dr. Gatel and collaborators provides some useful information for understanding of the magnetic phase transition in an FeRh film. The electron microscopy observations definitely show a variation of phase transition temperatures within the FeRh layer sandwiched by nonmagnetic substances. However, my opinion is that this study is still incomplete, because of significant technical concerns that need further examinations. Although the discussion using the magnetic induction maps provide beneficial information for the study of this particular system, I am nevertheless not sure whether this manuscript offers wide interests to readers of nature communications. This paper needs significant revisions to address the technical concerns and show the suitability for publishing in nature communications, otherwise should be published in a specialized journal of materials science without further delays.

- page 2: Although the sample thickness was roughly estimated at 90 nm, which was deduced from SEM images such as shown in Fig. S1, the sample prepared by using focused ion beam must be wedge-shaped: it is accordingly difficult to evaluate the true thickness in the portion of FeRh using the SEM images. When the sample thickness is unclear, it is difficult to discuss about the magnetic induction values.

We agree with the referee that the FIB specimen is wedge shaped. This affected mainly the thick Pt protection layer that has been removed to ensure an overlap between the reference area and the FeRh layer for holography experiments (see figure S1a). We then performed the thickness measurement of the lamella.

The FeRh layer, deeper in the lamella, has thus a nearly constant thickness of 90 ± 5 nm. The error bar has been determined by measuring the width of the lamella in many places. The error bar in the width measurement has been integrated in the magnetization data deduced from the phase maps.

- page 3: Electron holography must be sensitive to magnetic induction B_x , rather than magnetization M_x . The relationship of eq.2 is invalid unless the effect of leakage magnetic field, which must be present outside of the sample, is

incorporated correctly.
In the ferromagnetic state, the “wire” geometry of the FeRh in the lamella aligns the magnetization along the wire axis due to its strong shape anisotropy. In this state, the stray field can be considered negligible and the magnetization equals the induction. It is important to note that VSM experiments also measure induction that is considered equal to magnetization for the very same reasons. The equivalence between induction and magnetization is a usual assumption in magnetometry measurements. This has been explained just before equation 2 where the sentence « assuming that the magnetization is equivalent to the induction inside the FeRh layer » specifies the difference between induction and magnetization. The sentence has been modified : « assuming that the magnetization is equivalent to the induction inside the FeRh layer, i.e. neglecting the stray fields »
 page 3: “The contrast layers...are attributed to the presence of defects (dislocations...)”. As the authors indicate, structural imperfections appear to influence the magnetic phase transition significantly. The “presence of defects” should be discussed in greater detail.
As already stated, the AF phase is easy to destabilize by any structural defect (dislocations, vacancies, ...). Dislocations for instance induce strain fields that will break the compensation between Fe and Rh magnetic moments and will destroy locally the AF order. This is what also what is observed by Le Graët el al. when they introduce dopants and hence disorder in a FeRh layer.
 page 3: My naïve question is, since the layer of MgO is of insulator, it must be strongly electric-charged by electron illumination. However, Fig. 2d does not show an appreciable phase shift due to electric charging in the MgO layer. The result is not consistent with the insulating nature in the MgO layer. Provided that the phase shift due to electric charging was already removed from the curves shown in Fig. 2d, the slope in the FeRh area must be affected accordingly.
The MgO layer is covered by conducting materials (FeRh, ink and platinum) that enable the evacuation of the electric charges. Every phase shift map shows only the magnetic contribution. The electrostatic contribution (inner potential, ...) to the phase shift maps has been measured and removed as described in SI (paragraph d).
 page 4: It is unclear why the “antiferromagnetic” state shows a definite magnetization 0.28 T. Is this state “ferrimagnetic”?
The total magnetization is measured to be 0.28T but this does not mean that the material is ferrimagnetic as Fe₃O₄, for instance. The magnetization measured in the EH experiment is the sum of the contributions of all the layers crossed by the electrons along the electron beam path. As illustrated on figure S4, a 80nm-thick crystalline layer and two 5nm-thick amorphous layers are crossed by the electrons. At low temperatures (-120°C), most of the sample is an antiferromagnetic state however we still measure a small ferromagnetic component. That is mostly attributed in the text to both FIB damaged layers which remain in the ferromagnetic state. The total magnetization different from zero at low temperature is then explained by various contributions (amorphous layers, interfaces) and not by a ferrimagnetic behaviour of the FeRh crystalline layer.
 page 5: The authors ascribe the disagreements in the shape of M-T curves, shown in Fig.3, to the structural damage that may be induced by focused ion beam. To justify this picture, both the magnetization and depth of the damage layer should be determined accurately. The effect of structural disorder on the Neel/Curie temperatures in this

compound should be mentioned more deeply, in order to support the authors' conclusion.

- In addition, we would like more reasonable grounds as to why the temperature hysteresis and the transition point are reduced significantly in the foil sample

As in any techniques, a priori information must be used to interpret the results. In our case, such information concerns the estimated thickness damaged in a FIB preparation process. We have estimated that this depth is at least 5nm (commonly reported minimum depth) as function of FIB preparation conditions and this seems consistent with our results.

The magnetization of the whole FeRh layer has been measured at high temperature when in the FM state. We can assume that the magnetization of the damage layer is at least the same as low temperature, but more probably higher (Curie-Law behavior).

The reviewer points to the difficulty of comparing macroscopic and local properties. Indeed, the VSM results showing the magnetization versus temperature plot averaged in a 2mmx3mm area show differences with the results by EH measurements collected on a much smaller area (50nm x 90nm). However, the presented results are rather complementary, owing to the different scales of observation, than opposite. TEM related techniques are not the most suitable for collecting statistical information and are likely to emphasize micro/nanoscale inhomogeneities. However, as surprising as the observed differences may seem, they mainly concern the decrease of the transition temperature and the narrowing of the hysteresis cycle for the FIB prepared specimen.

The comparison with macroscopic temperature hysteresis and transition point for all the layer would be possible if a statistical analysis had been possible. The opening of the macroscopic M(T) cycle for example reflects the dispersion of the cycles for each area. In addition, the resulting magnetic coupling between 5nm-thick amorphous layers due to the FIB preparation and the remaining B2 phase acts as a strong external magnetic field which decreases the transition temperature as shown using VSM in many publications.

- page 7: What kind of “symmetry breaking” occurs in the FeRh sample.

The expression « symmetry breaking » is misleading and has been replaced by « the presence of interfaces ».

- page 7: “They correspond to the nucleation of FM areas.” I guess, a significant volume change (1%) associated with the ferromagnetic/antiferromagnetic transition provides undesired diffraction contrasts, and accordingly additional phase shift in electron holography data. This point should be addressed

The presence of diffraction will induce amplitude variations but no magnetic phase modification in the process of image formation. The volume change will therefore not affect the magnetic phase shift and the quantification of the magnetization. In any case, we avoided diffraction contrasts and never observed diffraction contrast in the heating/cooling process.

REVIEWERS' COMMENTS:

Reviewer #1 (Remarks to the Author):

I have read through the authors' response to my technical queries and I am satisfied that they have fully engaged with them and made the relevant modifications to the manuscript where appropriate. In my first report, I queried the wider generality of this work to the study of FeRh. I find that the authors have addressed this in the modifications to the manuscript presented. Therefore, I find the manuscript to be suitable for acceptance in Nat. Comms.

Reviewer #2 (Remarks to the Author):

I feel that the corrections made to the article are sufficient in addressing my concerns thus far. At this point i feel that the manuscript is clear, complete and ready for publication.

Your argument for keeping units in Tesla also make sense for the application to the theorem.

Reviewer #3 (Remarks to the Author):

The technical points raised by reviewer #3 have been addressed almost satisfactorily. The revised version of the manuscript appears to be suitable for publication.